# Identification of circulating miRNAs as fracture-related biomarkers

**Elena Della Bella**[1☯], **Ursula Menzel**[1☯], **Andreas Naros**[1,2], **Eva Johanna Kubosch**[3], **Mauro Alini**[1], **Martin J. Stoddart**[1,3]*

1 AO Research Institute Davos, Davos Platz, Switzerland, 2 Department of Oral and Maxillofacial Surgery, Tübingen University Hospital, Tübingen, Germany, 3 Department of Orthopedics and Trauma Surgery, Faculty of Medicine, Medical Center-Albert-Ludwigs-University of Freiburg, Albert-Ludwigs-University of Freiburg, Freiburg, Germany

☯ These authors contributed equally to this work.
* martin.stoddart@aofoundation.org

**Data Availability Statement:** Sequencing data deposited in NCBI's Gene Expression Omnibus with accession number GSE217879. The minimal

## Abstract

Fracture non-unions affect many patients worldwide, however, known risk factors alone do not predict individual risk. The identification of novel biomarkers is crucial for early diagnosis and timely patient treatment. This study focused on the identification of microRNA (miRNA) related to the process of fracture healing. Serum of fracture patients and healthy volunteers was screened by RNA sequencing to identify differentially expressed miRNA at various times after injury. The results were correlated to miRNA in the conditioned medium of human bone marrow mesenchymal stromal cells (BMSCs) during *in vitro* osteogenic differentiation. hsa-miR-1246, hsa-miR-335-5p, and miR-193a-5p were identified both *in vitro* and in fracture patients and their functional role in direct BMSC osteogenic differentiation was assessed. The results showed no influence of the downregulation of the three miRNAs during *in vitro* osteogenesis. However, miR-1246 may be involved in cell proliferation and recruitment of progenitor cells. Further studies should be performed to assess the role of these miRNA in other processes relevant to fracture healing.

## Introduction

Fracture non-unions affect many patients leading to functional and psychosocial disability [1]. Several risk factors have been described and include biological, surgical, and mechanical factors. However, none of them are able to determine if healing abnormalities will arise [1, 2]. As a consequence, there are no prognostic markers to monitor fracture healing, leaving no option but to wait until a non-union has occurred. Bone turnover markers have some utility but there are still uncertainties in their clinical use for monitoring fracture healing [3, 4]. The identification of new biomarkers, alone or in combination, revealing healing abnormalities at early stages are therefore crucial to improve non-union treatment or to prevent its occurrence.

miRNA are short nucleotide sequences [∼20–25 nt) that modulate gene expression at a post-transcriptional level, thereby regulating virtually all biological processes [5]. miRNAs are not only found in the intracellular space, but they are also secreted and can be easily detected

underlying data set is available as supplementary file.

**Funding:** MJS and EDB obtained funding from AO Foundation and AO Trauma. https://www. aofoundation.org/what-we-do/research-innovation/ about https://www.aofoundation.org/trauma The funders had no role in study design, data collection and analysis, decision to publish, or preparation of the manuscript.

**Competing interests:** The authors have declared that no competing interests exist.

in many biological fluids, including serum and plasma [6]. Endogenous circulating miRNA are usually protected from RNase-mediated degradation and have found to be dysregulated in a variety of pathological conditions [6, 7].

miRNA has been extensively implicated in the control of bone formation, remodelling, and disease [8–11]. For example, miRNAs have been shown to directly regulate osteoblast differentiation, including early regulation of the osteogenic factor Runx2 [8, 12], or to be associated with osteoporosis [13, 14]. In normal fracture healing, the haematoma miRNA signature was profiled, with some of the identified miRNAs being directly involved in healing processes [15]. Moreover, a profile of circulating miRNA could be associated to osteoporotic fractures, matching that of bone tissue samples from the same patients [14], proving that it is possible to sample the miRNome of the fracture callus at a distance with a minimally invasive procedure. In another study, a decrease in miR-92a was observed in healthy patients 24 hours after fracture [16], and specifically blocking this miRNA in preclinical models led to a more robust fracture repair [16].

Altogether, these studies suggest that circulating miRNA represent promising blood-based theranostic markers in bone disease and fracture healing [11].

Therefore, the aim of this study was to identify miRNA changes in the serum of fracture patients versus healthy controls, and correlate those changes with differential miRNA expression during *in vitro* osteogenic differentiation.

## Materials and methods

### Fracture patients and samples

Whole blood was collected with full ethical approval and signed informed patient consent (EK-Freiburg 105/17 and ZH BASEC-Nr.2017-01390) from 10 healthy volunteers and 12 fracture patients (**S1 Table**). From fracture patients, blood was collected at day 3 (n = 10, "early"), between day 5 and 12 (n = 5, "late"), and between day 19 and 56 (n = 6, "very late") after fracture. After clotting, samples were centrifuged for 10 min at 2500g and 4°C. The serum was then filtered and frozen at −80°C.

### BMSC isolation and expansion

Human BMSCs were isolated from vertebral bone marrow aspirates with informed consent and full ethical approval (Freiburg EK 135/14). BMSCs from 6 male donors were used (mean age 59±16 years, range 33–80 years) for the sequencing experiment (n = 3) and/or for validation studies (n = 4). One donor was shared between the sequencing experiment and the validation studies *in vitro*.

After density gradient centrifugation, mononucleated cells were collected and seeded in Minimum Essential Medium alpha (αMEM, Gibco, Thermo Fisher, Zürich, Switzerland) with 10% MSC-qualified foetal bovine serum (FBS, Pan Biotech, Aidenbach, Germany), 100 μg/mL streptomycin, 100 U/mL penicillin (Gibco), and 5 ng/ml basic fibroblast growth factor (bFGF, Fitzgerald Industries International, Acton, MA, USA). Non-adherent cells were removed after 4 days and BMSCs colonies were grown to 70–80% confluency. Cells were reseeded at a density of $3x10^3$ cells/cm$^2$ for expansion. Experiments were performed with BMSCs at passage 2–3.

### *In vitro* osteogenic differentiation

BMSCs (n = 3 donors) were seeded at a density of $1.5x10^4$ cells/cm$^2$ and committed to osteogenic differentiation using Dulbecco's Modified Eagle's Medium (DMEM) 1 g/L glucose (Gibco), 10% heat-inactivated FBS (Gibco), 100 μg/mL streptomycin, 100 U/mL penicillin, 5

mM β-glycerophosphate, 50 μg/ml L-Ascorbic acid 2-phosphate, and 10 nM water-soluble dexamethasone (Sigma-Aldrich) [17]. Undifferentiated controls (ctrl) were cultured in DMEM, 10% FBS and antibiotics only. Day 7 and day 14 conditioned medium samples were generated and used for subsequent analyses. After culturing cells for 48h in serum-free conditions, medium was collected, centrifuged at 2000g for 30 min, and snap frozen before storing at -80˚C. Medium was refreshed three times/week.

## miRNA sequencing

Medium (n = 3 ctrl and n = 3 osteo, for day 7 and day 14 timepoints) and serum samples were processed by Qiagen Genomic Services (Qiagen, Hilden, Germany) for miRNA sequencing and bioinformatics. RNA was isolated using the miRNeasy Serum/Plasma Kit and NGS library was built using the QIAseq miRNA Library Kit (Qiagen). UMI-containing adapters and PCR indices were added. Sequencing was performed on an Illumina NextSeq 500 sequencer (single end, 75 nt reads). Raw data was de-multiplexed and FASTQ files were generated using bcl2fastq (Illumina, San Diego, CA) and checked with FastQC. UMI and adapter information was extracted with Cutadapt (1.11). After adapter removal and UMI correction, Bowtie2 (2.2.2) was used to map reads against mirbase20 (reference genome: GRCh37). miRNA differential expression was analysed with EdgeR. miRNAs were considered as differentially expressed with a Log2FC>|0.58| (i.e., |1.5|-fold-change) and with a $p<0.05$. False Discovery Rate (FDR)-corrected p-values were calculated with the Benjamini-Hochberg method.

## miRNA inhibitor treatment

Control and osteogenic cultures, set up as described above (n = 4 donors), were treated with miRCURY LNA miRNA inhibitors for hsa-miR-193a-5p, hsa-miR-335-5p, and hsa-miR-1246, or with the Negative Control B (Qiagen). Ten picomoles of miRNA inhibitors or negative control were delivered using Lipofectamine™ RNAiMAX (Thermo Fisher), using a 1:3 RNA:Lipofectamine ratio. Cells were cultured up to 21 days to evaluate mineral deposition by Alizarin Red staining, while samples for RNA isolation and RT-qPCR were collected at day 2, 8 and 21 after transfection.

## Analysis of mineral deposition

Mineral deposition was evaluated by Alizarin Red staining. Cells were fixed with 10% buffered formalin and stained with 40 mM Alizarin Red (Sigma-Aldrich). Macroscopic images were captured using a Raspberry Pi Camera (Raspberry Pi Ltd, Cambridge, UK) on a custom 3D-printed structure. After imaging, the staining was quantified with the cetylpyridinium method [18].

## RNA isolation

miRNAs were isolated from conditioned medium using Qiagen miRNeasy Serum/Plasma kit. Following the miRNA inhibitor treatment experiment, total RNA isolation was performed as previously described [18] and used for both cDNA synthesis and miRNA reverse transcription. RNA concentration was determined using a Qubit 4.0 Fluorometer in combination with the appropriate Assay Kit: Qubit microRNA Assay Kit for conditioned medium miRNA and Qubit RNA HS Assay Kit for cellular total RNA (Thermo Fisher).

## Quantitative Real-Time PCR

cDNA synthesis for analysis of coding gene expression in BMSCs was performed using TaqMan Reverse Transcription reagents (Thermo Fisher). The expression of differentiation

markers and putative miRNA target genes (S2 Table) was analysed by qPCR as previously described [18]. Results were expressed as $2^{-\Delta Ct}$, with *RPLP0* as a reference gene. miRNA putative targets were selected with miRbase or TarBase v.8 search and literature sources [19–25] and included *BCL2*, *ICAM1*, *PDE1C*, *RUNX2* (S2 Table).

Reverse transcription of miRNA from BMSCs was carried out using the TaqMan™ Advanced miRNA cDNA Synthesis Kit (Thermo Fisher). The expression of hsa-miR-1246, hsa-miR-193a-5p, and hsa-miR-335-5p was assessed by qPCR using TaqMan Advanced miRNA Assays (see S2 Table for details) and TaqMan Fast Advanced Master Mix (Thermo Fisher).

Qiagen miRCURY LNA RT Kit was used for serum miRNA reverse transcription. Qiagen miRCURY LNA SYBR Green PCR Kit was used to set up miRCURY LNA miRNA Custom PCR panel plates for miRNA analysis (S2 Table). Results were calculated as $2^{-\Delta\Delta Ct}$, relative to the undifferentiated control and using global miRNA average as reference [26].

## Visualization and statistical analysis

Volcano plots and Venn diagram were realized using ggplot2 and ggvenn R packages. Graph-Pad Prism 9.3.1 was used for statistical analysis, using a repeated-measures two-way ANOVA with Tukey's multiple comparison test. For qPCR analysis of miRNA from conditioned medium a one-sample t-test was used to compare the miRNA average fold-change to the null hypothesis (fold-change = 1 = no differential regulation).

## Results

### miRNA signature in osteogenic conditioned medium from human BMSCs

In day 7 medium, sequencing identified 74 downregulated and 46 upregulated miRNAs in osteogenic differentiation vs. undifferentiated controls (Fig 1A). Table 1 reports the 25 most significant miRNAs, while S3 Table includes all differentially expressed miRNAs. On day 7, miR-362-5p, miR-136-5p, miR-374a-3p, miR-3065-5p, miR-942-5p were not detected during osteogenic differentiation, while being detectable in undifferentiated cells with low TMM values (S3 Table). qPCR analysis confirmed the downregulation of hsa-miR-31-5p, hsa-miR-125a-5p, hsa-miR-335-5p, hsa-miR-23b-3p, hsa-miR-411-5p, hsa-miR-26b-5p, and hsa-miR-103a-3p, and the upregulation of hsa-miR-7704, hsa-miR-193a-5p, and hsa-miR-423-3p (Fig 1B).

On day 14, 19 miRNAs were downregulated and 25 upregulated in osteogenic differentiation vs. undifferentiated controls (S1 Fig). Table 2 reports the 25 most significantly differentially expressed miRNAs, and S4 Table the full list of differentially expressed miRNAs.

As the most significant changes in miRNA expression were identified within the first week of differentiation, the following analyses using *in vitro* data focused on day 7 miRNAs only.

### miRNA signature in serum from fracture patients

When comparing all serum samples from fracture patients with healthy volunteers, 19 miRNAs were downregulated and 45 upregulated (Fig 2).

Day 0–3 serum samples from fracture patients versus controls revealed 21 downregulated and 38 upregulated miRNAs (S2A Fig). The comparison between samples 5–12 days post fracture vs controls identified 32 downregulated and 38 upregulated miRNAs (S2B Fig). Finally, 26 miRNAs were downregulated and 20 were upregulated between day 19–56 samples vs controls (S2C Fig). Tables 3–6 report the 25 most significantly differentially expressed miRNAs for all the comparisons.

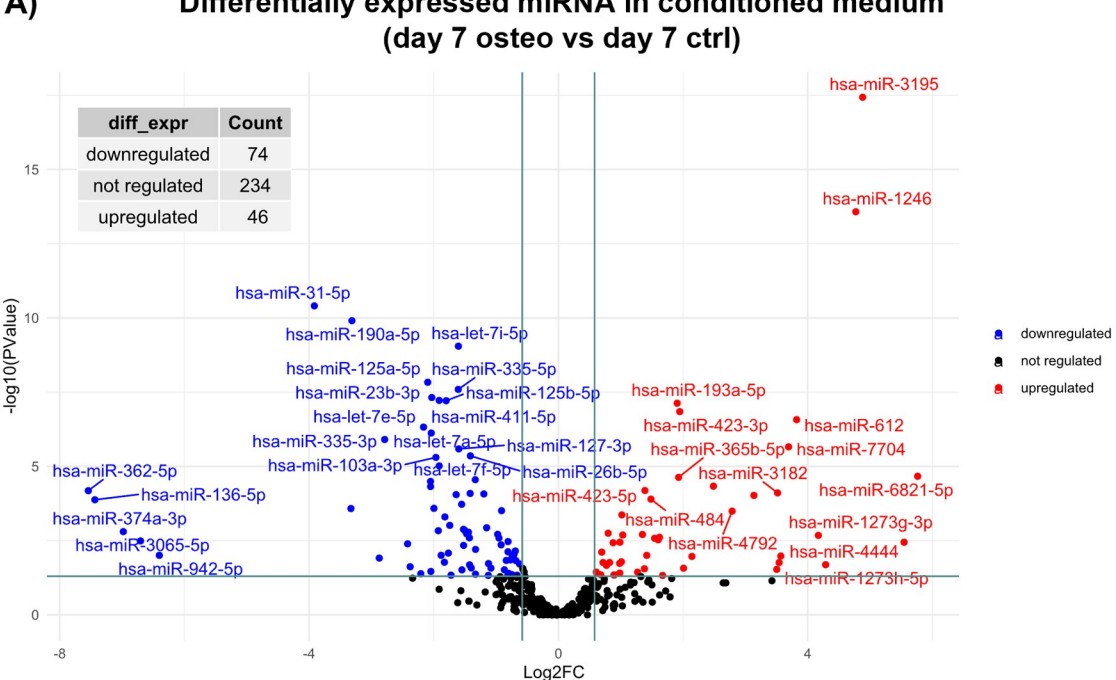

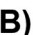

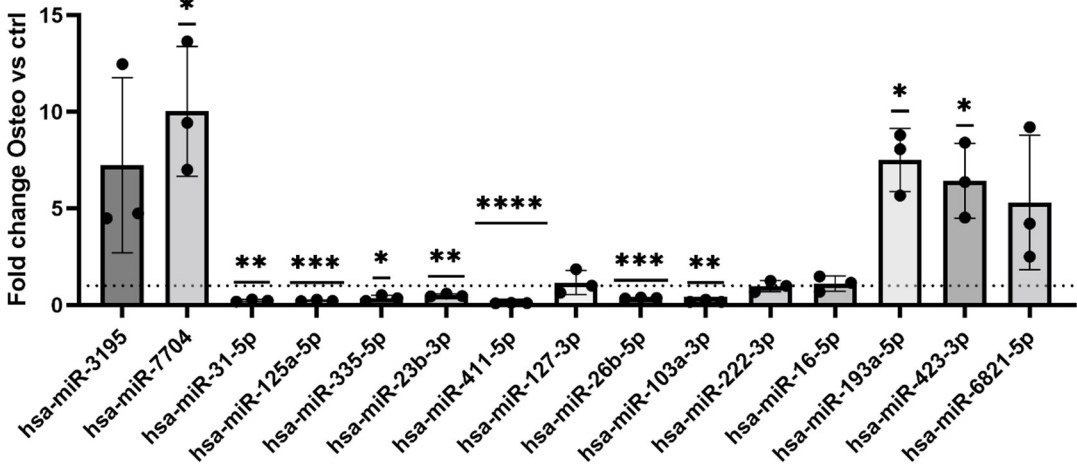

**Fig 1. Analysis of differentially expressed miRNA in conditioned medium.** In volcano plots, the x-axis reports the log2 fold-change between osteo and ctrl (Log2FC), while the y-axis represents the -log10 of the p-value. Thresholds are set at 0.58 of log2FC (corresponding to an absolute 1.5-fold-change value) and to a -log10 p-value of 1.3 (corresponding to p-value of 0.05). A) day 7 osteo vs ctrl volcano plot. B) qPCR validation of a selection of differentially expressed miRNAs at day 7. Data are expressed as $2^{-\Delta\Delta Ct}$, comparing osteo vs. CRL and normalizing individual miRNA expression to the miRNA global average. *$p<0.05$; **$p<0.01$; ***$p<0.001$; ****$p<0.0001$. Dotted line is set at 1, which represent for each miRNA the level of the control (undifferentiated) sample.

**Table 1. List of top 25 differentially regulated miRNA at day 7 in conditioned medium (osteo vs ctrl).** Average Trimmed Mean of M (TMM) values for each group are reported. miRNAs are sorted by False Discovery Rate (FDR).

| miRNA | Log2FC | PValue | FDR | Osteo average TMM | Ctrl average TMM |
|---|---|---|---|---|---|
| hsa-miR-3195 | 4.881 | 3.68E-18 | 1.30E-15 | 278.79 | 9.31 |
| hsa-miR-1246 | 4.771 | 2.65E-14 | 4.69E-12 | 400.67 | 14.47 |
| hsa-miR-31-5p | -3.916 | 3.92E-11 | 4.63E-09 | 119.14 | 2012.86 |
| hsa-miR-190a-5p | -3.313 | 1.24E-10 | 1.10E-08 | 31.33 | 318.82 |
| hsa-let-7i-5p | -1.604 | 8.93E-10 | 6.32E-08 | 31695.58 | 96381.93 |
| hsa-miR-125a-5p | -2.097 | 1.47E-08 | 8.65E-07 | 4300.11 | 18439.02 |
| hsa-miR-335-5p | -1.607 | 2.55E-08 | 1.29E-06 | 573.95 | 1777.66 |
| hsa-miR-23b-3p | -2.031 | 4.78E-08 | 2.12E-06 | 627.78 | 2630.9 |
| hsa-miR-411-5p | -1.912 | 5.99E-08 | 2.18E-06 | 165.17 | 659.65 |
| hsa-miR-125b-5p | -1.801 | 6.15E-08 | 2.18E-06 | 24417.32 | 85125.36 |
| hsa-miR-193a-5p | 1.906 | 7.45E-08 | 2.40E-06 | 1037.99 | 272.71 |
| hsa-miR-423-3p | 1.947 | 1.43E-07 | 4.23E-06 | 2302.71 | 593.04 |
| hsa-miR-612 | 3.822 | 2.67E-07 | 7.26E-06 | 614.75 | 44.31 |
| hsa-let-7e-5p | -2.164 | 4.76E-07 | 1.20E-05 | 3312.79 | 14921.88 |
| hsa-let-7a-5p | -2.040 | 7.51E-07 | 1.77E-05 | 44369.19 | 182487.8 |
| hsa-miR-335-3p | -2.787 | 1.24E-06 | 2.75E-05 | 12.8 | 132.79 |
| hsa-miR-7704 | 3.693 | 2.19E-06 | 4.57E-05 | 111.34 | 8.51 |
| hsa-miR-127-3p | -1.598 | 2.57E-06 | 5.06E-05 | 470.54 | 1450.73 |
| hsa-miR-26b-5p | -1.412 | 4.42E-06 | 8.23E-05 | 2351.26 | 6304.47 |
| hsa-miR-103a-3p | -1.965 | 4.98E-06 | 8.82E-05 | 1645.22 | 6495.55 |
| hsa-let-7f-5p | -1.912 | 9.65E-06 | 1.63E-04 | 23577.43 | 88808.11 |
| hsa-miR-6821-5p | 5.763 | 2.15E-05 | 3.46E-04 | 51.6 | 0.59 |
| hsa-miR-365b-5p | 1.929 | 2.34E-05 | 3.60E-04 | 218.67 | 57.86 |
| hsa-miR-222-3p | -1.334 | 2.81E-05 | 4.14E-04 | 366.15 | 915.92 |
| hsa-miR-181a-2-3p | -2.052 | 3.21E-05 | 4.54E-04 | 137.38 | 626.84 |

## Identification of common miRNAs in fracture healing and *in vitro* osteogenesis

The most significant changes were identified early after osteogenic differentiation induction or during fracture healing, when common differentially expressed miRNAs were identified (**Fig 3**). miR-1246 and miR-193a-5p were upregulated both in early osteogenic differentiation *in vitro* and in fracture healing, while miR-335-5p showed an opposite direction of regulation, being downregulated during *in vitro* osteogenesis, and upregulated after fracture.

## Functional validation of miRNA candidates *in vitro* by miRNA inhibitor treatment

The functional role of miR-1246, miR-335-5p, and miR-193a-5p was assessed during *in vitro* osteogenesis, using inhibitors to repress miRNA function early during differentiation. Their expression was evaluated in non-transfected cells to further confirm their differential expression during osteogenic differentiation (**S3 Fig**). The expression of miR-1246 was non-significantly reduced at 2 days, while it was significantly upregulated at 21 days of osteogenic differentiation, compared to the controls. miR-335-5p was significantly downregulated in osteogenic differentiation at both timepoints, while miR-193a-5p did not show any significant difference between conditions.

**Table 2. List of top 25 differentially regulated miRNA at day 14 in conditioned medium (osteo vs ctrl).** Average Trimmed Mean of M (TMM) values for each group are reported. miRNAs are sorted by False Discovery Rate (FDR). Only 4 miRNAs could be identified with an FDR < 0.05.

| miRNA | Log2FC | PValue | FDR | Osteo average TMM | Ctrl average TMM |
|---|---|---|---|---|---|
| hsa-miR-335-5p | -1.640 | 2.19E-05 | 9.68E-03 | 377.16 | 1176.87 |
| hsa-miR-223-3p | -8.727 | 6.55E-05 | 1.45E-02 | 13.81 | 6343.38 |
| hsa-miR-199b-5p | 1.821 | 3.23E-04 | 3.69E-02 | 342.87 | 94.9 |
| hsa-miR-378a-3p | 1.391 | 3.34E-04 | 3.69E-02 | 832.03 | 314.7 |
| hsa-miR-142-5p | -7.400 | 1.03E-03 | 9.14E-02 | 0 | 191.48 |
| hsa-miR-30a-5p | 1.005 | 3.13E-03 | 2.30E-01 | 1557.25 | 775.13 |
| hsa-miR-146a-5p | 1.244 | 3.99E-03 | 2.39E-01 | 742.1 | 313.85 |
| hsa-miR-29a-3p | 0.956 | 4.32E-03 | 2.39E-01 | 10966.81 | 5651.41 |
| hsa-miR-3180-3p | -3.422 | 5.86E-03 | 2.68E-01 | 7.31 | 86.25 |
| hsa-miR-138-1-3p | 3.006 | 6.30E-03 | 2.68E-01 | 52.95 | 5.77 |
| hsa-miR-205-5p | -3.919 | 6.66E-03 | 2.68E-01 | 4 | 68.35 |
| hsa-miR-23b-5p | -3.068 | 8.90E-03 | 3.24E-01 | 3.31 | 35.91 |
| hsa-miR-4792 | 1.570 | 9.53E-03 | 3.24E-01 | 159.78 | 52.63 |
| hsa-miR-320d | 1.117 | 1.03E-02 | 3.24E-01 | 690.88 | 318.56 |
| hsa-miR-99a-5p | 1.006 | 1.36E-02 | 3.55E-01 | 1506.58 | 750.16 |
| hsa-miR-1228-3p | -4.488 | 1.37E-02 | 3.55E-01 | 0 | 24.21 |
| hsa-miR-30a-3p | 0.817 | 1.40E-02 | 3.55E-01 | 3348.24 | 1899.07 |
| hsa-miR-26a-2-3p | -4.449 | 1.45E-02 | 3.55E-01 | 0 | 22.18 |
| hsa-miR-125b-5p | -0.897 | 1.75E-02 | 3.99E-01 | 35619.39 | 66314.06 |
| hsa-miR-29b-3p | 1.052 | 1.80E-02 | 3.99E-01 | 1130.62 | 544.26 |
| hsa-miR-1246 | 1.224 | 1.92E-02 | 4.00E-01 | 639.78 | 272.02 |
| hsa-miR-34a-5p | 0.797 | 1.99E-02 | 4.00E-01 | 2810.69 | 1617.6 |
| hsa-miR-142-3p | -3.907 | 2.28E-02 | 4.24E-01 | 10.38 | 170.41 |
| hsa-miR-619-5p | 1.578 | 2.34E-02 | 4.24E-01 | 103.53 | 34.2 |
| hsa-miR-1262 | 1.524 | 2.57E-02 | 4.24E-01 | 78.75 | 25.14 |

Alizarin Red staining showed no differences in mineral deposition after treatment with miRNA inhibitors (**Fig 4**). The expression of early differentiation markers also did not change among treatment groups (**Fig 5**), while the expression of *SPP1* was significantly increased in cells treated with the miR-193a-5p inhibitor in comparison to the negative control (**Fig 6**). The levels of miR-1246, miR-335-5p and miR-193a-5p at any timepoint did not show any significant correlation to other osteogenic marker (data not shown).

The analysis of putative miRNA targets revealed no difference between the groups, with only *BCL2* showing an upregulation trend in osteogenic medium following miR-1246 inhibition (**Fig 7**). Data not shown for *TIMP2*, *ACVR2A*, *ACVR2B*, *SMURF2*, *CTNNB1*, and *BAX* (no differences among the miRNA inhibitor treatment groups).

## Discussion

The identification of biomarkers to stratify patients' non-union risk is crucial to allow early intervention. Blood-based biomarkers would be ideal since they can be obtained following a minimally invasive procedure and can be easily combined with the analysis of other biochemical parameters. To be robust, the marker must be present in a broad range of patients, for this reason we utilized an unmatched patient population. Despite this, we successfully identified potential markers and more clearly defined the time period when sample present the greatest number of miRNA changes.

## Differentially expressed miRNA in serum samples
## (fracture patients vs healthy controls)

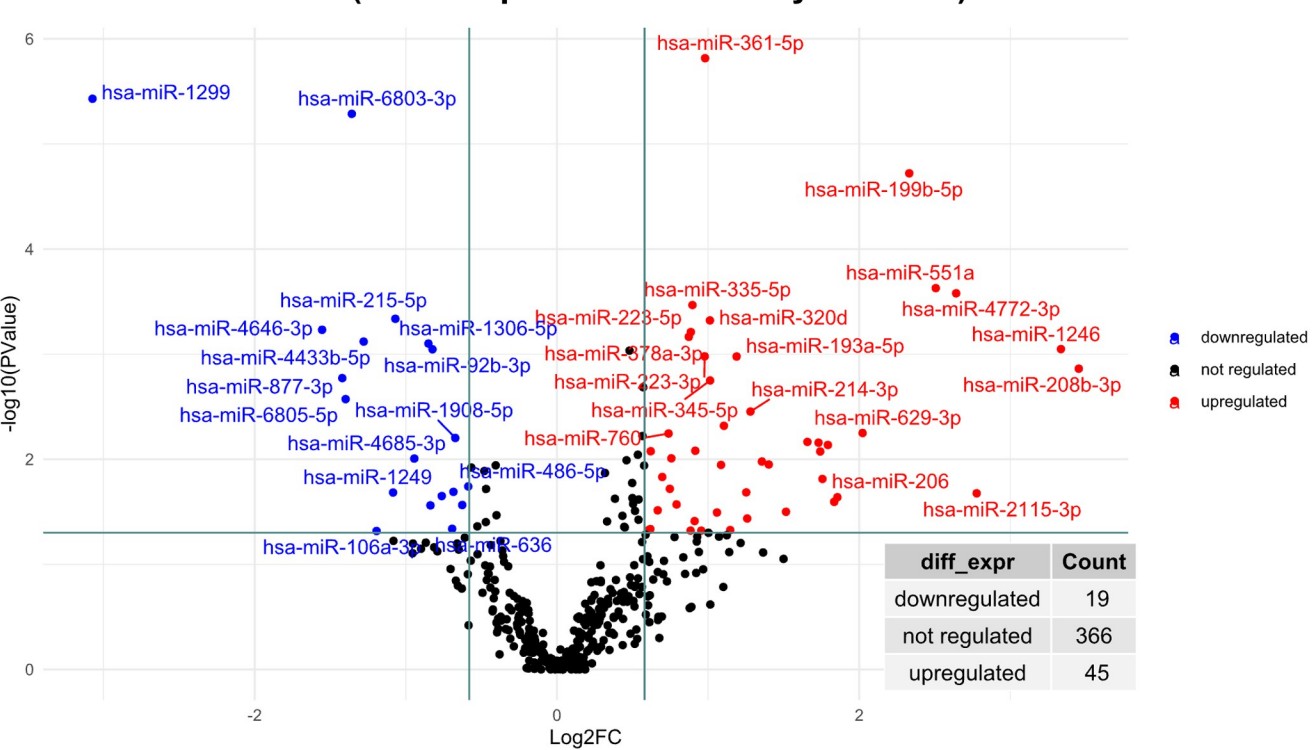

**Fig 2. Volcano plot identifying differentially expressed miRNAs in serum of fracture patients (all timepoints) versus controls.**

This study identified miRNAs in the serum of fracture patients or secreted during *in vitro* osteogenic differentiation that may play a physiological role in fracture healing. Within the patients' serum, several miRNAs were regulated within the first 12 days post-injury, and this number was reduced from 19 days onwards. It is likely that the release of fracture-related miRNA into the bloodstream is reduced once the vasculature has recovered and maintains its integrity. miRNA changes during *in vitro* osteogenic differentiation were also more substantial at early timepoints. Taken together, the identification of early markers of fracture appears feasible. Three miRNAs were selected for functional validation during *in vitro* BMSCs osteogenic differentiation. However, the inhibition of these miRNAs at the start of differentiation did not seemingly affect this process.

Due to the complexity of fracture healing, it is unlikely a single marker will be predictive, but a combination of multiple markers may offer a route to diagnosis. This might also be the reason behind the lack of observable changes with a single miRNA knockdown in this case.

hsa-miR-1246 was upregulated in the serum of fracture patients and in the conditioned medium of BMSCs undergoing osteogenic differentiation. hsa-miR-1246 is a positive regulator of proliferation and associated to cancer progression and malignancy [27–30]. miR-1246 targets negative regulators of cell cycle progression, such as cyclin G2 (*CCNG2*) [28, 30, 31]. Moreover, miR-1246 is a mechanosensitive miRNA, upregulated in periodontal ligament stem cells by stretch [32] or by compression [33]. In BMSCs, miR-1246 was found to be upregulated by SDF1/CXCL12, an important chemokine during fracture healing [34], and downregulated after anchorage loss [35], supporting a major role of miRNA in proliferation and cell survival.

**Table 3. List of 25 most differentially expressed known microRNAs in fracture serum samples (all timepoints) versus controls.** miRNAs are sorted by False Discovery Rate (FDR). In total, 24 miRNAs that were significantly differentially expressed at a significance level of 0.05 were identified (FDR).

| miRNA | Log2FC | PValue | FDR | Fracture average TMM | Control average TMM |
|---|---|---|---|---|---|
| hsa-miR-361-5p | 0.980 | 1.53E-06 | 6.57E-04 | 681.51 | 345.39 |
| hsa-miR-1299 | -3.073 | 3.71E-06 | 7.40E-04 | 3.01 | 27.89 |
| hsa-miR-6803-3p | -1.357 | 5.16E-06 | 7.40E-04 | 13.5 | 33.26 |
| hsa-miR-199b-5p | 2.331 | 1.90E-05 | 2.05E-03 | 69.69 | 13.86 |
| hsa-miR-551a | 2.507 | 2.35E-04 | 1.89E-02 | 5.67 | 0.76 |
| hsa-miR-4772-3p | 2.642 | 2.63E-04 | 1.89E-02 | 9.6 | 1.34 |
| hsa-miR-335-5p | 0.897 | 3.41E-04 | 2.09E-02 | 457.67 | 246.61 |
| hsa-miR-215-5p | -1.069 | 4.61E-04 | 2.28E-02 | 13.62 | 29.43 |
| hsa-miR-320d | 1.012 | 4.77E-04 | 2.28E-02 | 65.01 | 32.21 |
| hsa-miR-4646-3p | -1.554 | 5.86E-04 | 2.34E-02 | 1.63 | 6.38 |
| hsa-miR-223-5p | 0.886 | 6.16E-04 | 2.34E-02 | 581.59 | 316.22 |
| hsa-miR-378a-3p | 0.873 | 6.84E-04 | 2.34E-02 | 259.62 | 141.06 |
| hsa-miR-4433b-5p | -1.279 | 7.59E-04 | 2.34E-02 | 677.23 | 1642.52 |
| hsa-miR-1306-5p | -0.850 | 7.93E-04 | 2.34E-02 | 167.97 | 301.17 |
| hsa-miR-1246 | 3.335 | 8.95E-04 | 2.34E-02 | 82.43 | 8.08 |
| hsa-miR-92b-3p | -0.823 | 9.00E-04 | 2.34E-02 | 131.71 | 231.44 |
| hsa-miR-361-3p | 0.480 | 9.25E-04 | 2.34E-02 | 477.96 | 340.41 |
| hsa-miR-223-3p | 0.978 | 1.05E-03 | 2.38E-02 | 63713.21 | 32354.91 |
| hsa-miR-193a-5p | 1.189 | 1.05E-03 | 2.38E-02 | 548.66 | 240.77 |
| hsa-miR-208b-3p | 3.453 | 1.37E-03 | 2.95E-02 | 2.31 | 0 |
| hsa-miR-877-3p | -1.420 | 1.69E-03 | 3.46E-02 | 3.37 | 9.09 |
| hsa-miR-345-5p | 1.014 | 1.78E-03 | 3.48E-02 | 151.16 | 75.08 |
| hsa-miR-191-5p | 0.571 | 2.07E-03 | 3.86E-02 | 14245.17 | 9593.18 |
| hsa-miR-6805-5p | -1.398 | 2.68E-03 | 4.80E-02 | 2.58 | 6.66 |
| hsa-miR-214-3p | 1.280 | 3.52E-03 | 6.05E-02 | 11.28 | 4.17 |

miR-1246 was also reported to activate Wnt/β-catenin signalling by suppressing the expression of *AXIN2* and *GSK3B* [36].

A few reports in literature suggest a role for hsa-miR-1246 as a regulator of osteoblast and osteoclast formation. Zhou *et al.* reported that miR-1246, contained in FBS-derived exosomes, can attenuate adipogenic differentiation of human BM-MSCs [37], therefore having an indirect positive role on bone formation, while Nguyen et al. [38] found miR-1246 as significantly reduced in pagetic overactive osteoclasts. In another study, miR-1246 was the most upregulated miRNA in circulating EVs from osteoporotic patients compared to those from healthy controls [39]. The treatment of osteoclast precursors with miR-1246 mimic enhanced osteoclastogenesis (in accordance with Liao et al [40] which reported the activation of *NFATC1* downstream of mir-1246), but also upregulated *SP7* in osteoblasts, indicating a role in bone remodelling processes. However, the same authors observed a decreased expression of this miRNA in osteoblasts compared to undifferentiated MSCs. This, together with our data from differentiating MSCs, might suggest that miR-1246 expression peaks during the pre-osteoblast stage.

Though this miRNA does not appear to influence direct osteogenic differentiation of BMSCs, its role in fracture healing could be correlated to early healing processes involving proliferation and recruitment of progenitor cells. It remains to be investigated whether an altered expression of miR-1246 in the serum of fracture patients might predict healing disturbances.

**Table 4. List of 25 most differentially expressed known microRNAs in day 3 ("early") fracture serum samples versus controls.** miRNAs are sorted by False Discovery Rate (FDR). In total, 5 miRNAs that were significantly differentially expressed at a significance level of 0.05 were identified (FDR).

| miRNA | Log2FC | PValue | FDR | "early" Fracture average TMM | Control average TMM |
|---|---|---|---|---|---|
| hsa-miR-199b-5p | 2.693 | 5.34E-06 | 2.16E-03 | 88.75 | 13.66 |
| hsa-miR-361-5p | 0.891 | 2.06E-05 | 4.16E-03 | 635.09 | 341.63 |
| hsa-miR-1246 | 4.218 | 4.63E-05 | 6.04E-03 | 150.32 | 7.96 |
| hsa-miR-133a-3p | 2.014 | 6.09E-05 | 6.04E-03 | 120.09 | 29.72 |
| hsa-miR-4772-3p | 2.985 | 7.47E-05 | 6.04E-03 | 12.28 | 1.33 |
| hsa-miR-6803-3p | -1.376 | 2.20E-04 | 1.35E-02 | 13.16 | 32.86 |
| hsa-miR-4433b-5p | -1.412 | 2.34E-04 | 1.35E-02 | 599.07 | 1594.74 |
| hsa-miR-1306-5p | -0.980 | 4.70E-04 | 2.37E-02 | 149.66 | 296.22 |
| hsa-miR-1249 | -1.645 | 6.54E-04 | 2.94E-02 | 3.84 | 12.8 |
| hsa-miR-877-3p | -1.812 | 8.14E-04 | 3.29E-02 | 2.17 | 8.87 |
| hsa-miR-551a | 2.516 | 9.33E-04 | 3.43E-02 | 5.63 | 0.75 |
| hsa-let-7c-5p | -0.736 | 1.03E-03 | 3.48E-02 | 453.79 | 753.96 |
| hsa-miR-223-3p | 1.046 | 1.18E-03 | 3.56E-02 | 65763.53 | 31861.02 |
| hsa-miR-223-5p | 0.913 | 1.23E-03 | 3.56E-02 | 585.44 | 312.15 |
| hsa-miR-345-5p | 1.093 | 1.37E-03 | 3.69E-02 | 158.72 | 74.44 |
| hsa-miR-582-3p | 2.155 | 2.02E-03 | 5.10E-02 | 15.87 | 4.3 |
| hsa-miR-193a-5p | 1.161 | 2.19E-03 | 5.21E-02 | 533.32 | 238.27 |
| hsa-miR-1299 | -2.854 | 2.52E-03 | 5.66E-02 | 3.5 | 26.45 |
| hsa-miR-206 | 2.279 | 3.55E-03 | 7.03E-02 | 171.72 | 35.2 |
| hsa-miR-3613-3p | 1.225 | 3.63E-03 | 7.03E-02 | 10.2 | 3.59 |
| hsa-miR-6747-3p | -1.110 | 3.65E-03 | 7.03E-02 | 5.29 | 12.8 |
| hsa-miR-1304-3p | 0.865 | 3.95E-03 | 7.25E-02 | 26.78 | 13.64 |
| hsa-miR-335-5p | 0.726 | 5.38E-03 | 9.45E-02 | 400.06 | 242.44 |
| hsa-miR-542-3p | 1.072 | 5.76E-03 | 9.55E-02 | 17.41 | 8.38 |
| hsa-miR-215-5p | -1.082 | 5.91E-03 | 9.55E-02 | 13.64 | 29.04 |

hsa-miR-193a-5p was also upregulated in early fracture healing and during *in vitro* osteogenic differentiation. miR-193a-5p was previously reported to inhibit osteogenic differentiation of BMSCs [41] and to be downregulated during dexamethasone-induced osteogenic differentiation of BMSCs and human adipose derived stem cells [41, 42]. However, the review by Izadpanah *et al.* [43] discusses the role of miR-193a in osteosarcoma, where it is downregulated compared to healthy tissues, suggesting that a lower degree of differentiation can be associated to lower levels of miR-193a. In osteosarcoma, miR-193a-3p and miR-193a-5p influenced TGF-β signalling and other key pathways [44]. Intriguingly, miR-193a was also described as mechanosensitive, as its downregulation could improve topographical feature-induced osteogenic differentiation in absence of dexamethasone [45].

Finally, miR-335-5p was upregulated in fracture patients while its secretion from cells decreased. In a previous study, we already identified intracellular miR-335-5p as being downregulated in osteogenic differentiation at day 7 [46]. Similarly, miR-335-5p was less expressed during osteogenic differentiation of human MSCs and targeted *RUNX2* [47]. Conversely, miR-335-5p inhibited Wnt antagonist *DKK1* in murine cells [48], and its overexpression in mice promoted bone formation and regeneration [49]. These contrasting results between human and murine cells raise the question whether this miRNA might have a different role in different species [50].

**Table 5. List of 25 most differentially expressed known microRNAs in day 5–12 ("late") fracture serum samples versus controls.** miRNAs are sorted by False Discovery Rate (FDR). In total, 25 miRNAs that were significantly differentially expressed at a significance level of 0.05 were identified (FDR).

| miRNA | Log2FC | PValue | FDR | "late" fracture average TMM | Control average TMM |
|---|---|---|---|---|---|
| hsa-miR-335-5p | 1.351 | 2.41E-06 | 9.46E-04 | 592.66 | 233.69 |
| hsa-miR-361-5p | 1.261 | 7.73E-06 | 1.39E-03 | 791.15 | 330.15 |
| hsa-miR-193a-5p | 1.633 | 1.06E-05 | 1.39E-03 | 713.3 | 230.23 |
| hsa-miR-320d | 1.607 | 1.52E-05 | 1.50E-03 | 94.13 | 31.07 |
| hsa-miR-208b-3p | 4.557 | 5.03E-05 | 3.95E-03 | 5.32 | 0 |
| hsa-miR-320c | 1.160 | 6.43E-05 | 4.21E-03 | 240.95 | 107.75 |
| hsa-miR-320b | 1.055 | 1.06E-04 | 5.93E-03 | 311.08 | 149.25 |
| hsa-miR-345-5p | 1.189 | 3.72E-04 | 1.83E-02 | 163.3 | 71.89 |
| hsa-miR-214-3p | 1.768 | 5.48E-04 | 2.21E-02 | 15.59 | 3.91 |
| hsa-miR-127-3p | -1.705 | 5.76E-04 | 2.21E-02 | 11.18 | 39.74 |
| hsa-miR-320a | 1.083 | 6.18E-04 | 2.21E-02 | 5673.54 | 2677.84 |
| hsa-miR-199b-5p | 1.735 | 7.79E-04 | 2.55E-02 | 42.83 | 13.12 |
| hsa-miR-370-3p | -1.513 | 9.81E-04 | 2.97E-02 | 28.49 | 83.9 |
| hsa-miR-191-5p | 0.767 | 1.10E-03 | 2.98E-02 | 15652.75 | 9198.25 |
| hsa-miR-223-3p | 1.036 | 1.14E-03 | 2.98E-02 | 62920.52 | 30690.45 |
| hsa-miR-223-5p | 0.999 | 1.38E-03 | 3.40E-02 | 597.1 | 300.34 |
| hsa-miR-381-3p | -1.666 | 1.47E-03 | 3.40E-02 | 10.96 | 33.93 |
| hsa-miR-483-5p | 1.690 | 1.60E-03 | 3.50E-02 | 771.66 | 238.93 |
| hsa-miR-409-3p | -1.346 | 1.78E-03 | 3.68E-02 | 259.68 | 660.73 |
| hsa-miR-342-5p | -1.547 | 2.07E-03 | 4.08E-02 | 2.33 | 9.75 |
| hsa-miR-485-3p | -1.654 | 2.37E-03 | 4.18E-02 | 77.01 | 236.24 |
| hsa-miR-378a-3p | 1.063 | 2.42E-03 | 4.18E-02 | 287.48 | 137.24 |
| hsa-miR-4646-3p | -2.839 | 2.45E-03 | 4.18E-02 | 0.52 | 6.21 |
| hsa-miR-136-3p | -2.282 | 2.91E-03 | 4.77E-02 | 1.02 | 6.63 |
| hsa-miR-133a-3p | 2.015 | 3.07E-03 | 4.83E-02 | 117.23 | 28.79 |

Altogether, our results suggest that miR-1246, miR-193a-5p, and miR-335-5p are detectable in the serum of fracture patients with reproducible patterns of expression and show a differential expression during in vitro osteogenic differentiation of human BMSC, but they are not directly causative of direct ossification. Therefore, we hypothesise that this is miRNA pattern is a downstream change, rather than an upstream regulator. Notwithstanding, we cannot exclude that these miRNAs have a functional role in other fracture healing-related processes. The effect of miRNA inhibition was studied only during direct *in vitro* osteogenic differentiation. This process is neither the only nor the main contributor to fracture healing. Further studies should be focused on validating their role during endochondral ossification or in other relevant mechanisms contributing to healing, such as angiogenesis. One limitation of this study is represented by the small cross-sectional cohort of patients recruited for this study, with no known fracture healing outcome. This limits the power of our findings; nonetheless, our results contribute to the identification of miRNA that can be further analysed and validated in larger longitudinal studies. While the clinical study included both biologically male and female fracture patients, the healthy controls and the *in vitro* study included only male donors. However, it is mandatory to extend the investigations to more female donors and clinical studies should take the biological sex into account as a variable, as there are several studies indicating potential sexual dimorphism in miRNA expression [51–54]. Whether those differences can be also identified *in vitro*, where the hormonal differences are neglected, is worth of investigation.

**Table 6. List of 25 most differentially expressed known microRNAs in day 19–56 ("very late") fracture serum samples versus controls.** miRNAs are sorted by False Discovery Rate (FDR). In total, only 1 miRNA that was significantly differentially expressed at a significance level of 0.05 was identified (FDR).

| miRNA | Log2FC | PValue | FDR | "very late" fracture average TMM | Control average TMM |
|---|---|---|---|---|---|
| hsa-miR-6803-3p | -1.630 | 5.52E-06 | 2.12E-03 | 10.43 | 33.42 |
| hsa-miR-361-5p | 0.658 | 4.37E-04 | 8.41E-02 | 548.35 | 347.17 |
| hsa-miR-199b-5p | 1.670 | 6.73E-04 | 8.64E-02 | 44.12 | 14.1 |
| hsa-miR-103a-3p | 0.604 | 1.14E-03 | 1.10E-01 | 7267.5 | 4780.48 |
| hsa-miR-92b-3p | -1.015 | 1.53E-03 | 1.10E-01 | 117.55 | 233.15 |
| hsa-miR-106b-5p | -1.031 | 2.34E-03 | 1.10E-01 | 19.15 | 38.48 |
| hsa-miR-144-5p | 0.924 | 2.80E-03 | 1.10E-01 | 402.56 | 209.5 |
| hsa-miR-3688-3p | -1.848 | 2.86E-03 | 1.10E-01 | 2.42 | 9.2 |
| hsa-miR-1294 | -1.168 | 2.90E-03 | 1.10E-01 | 23.63 | 50.76 |
| hsa-miR-1306-5p | -0.809 | 3.47E-03 | 1.10E-01 | 172.13 | 302.99 |
| hsa-miR-6511b-3p | -1.449 | 3.76E-03 | 1.10E-01 | 4.77 | 15.21 |
| hsa-miR-6511a-3p | -1.455 | 3.77E-03 | 1.10E-01 | 5.1 | 13.65 |
| hsa-miR-191-5p | 0.499 | 3.79E-03 | 1.10E-01 | 13678.55 | 9683.56 |
| hsa-miR-3605-3p | -0.862 | 4.21E-03 | 1.10E-01 | 29.96 | 53.88 |
| hsa-miR-1299 | -3.306 | 4.31E-03 | 1.10E-01 | 2.24 | 26.37 |
| hsa-miR-215-5p | -1.150 | 4.56E-03 | 1.10E-01 | 12.25 | 29.89 |
| hsa-miR-4772-3p | 1.858 | 5.99E-03 | 1.36E-01 | 5.63 | 1.37 |
| hsa-miR-125b-2-3p | 1.381 | 6.63E-03 | 1.42E-01 | 12.29 | 4.72 |
| hsa-miR-3614-5p | 1.662 | 8.04E-03 | 1.55E-01 | 16.4 | 6.34 |
| hsa-miR-339-5p | 0.542 | 8.69E-03 | 1.55E-01 | 659.46 | 453.45 |
| hsa-miR-3913-5p | -1.794 | 8.73E-03 | 1.55E-01 | 2.7 | 7.95 |
| hsa-miR-296-5p | -1.032 | 8.86E-03 | 1.55E-01 | 17.96 | 36.99 |
| hsa-miR-17-5p | -0.563 | 1.00E-02 | 1.68E-01 | 226.38 | 332.16 |
| hsa-miR-486-5p | -0.846 | 1.09E-02 | 1.75E-01 | 145924 | 262322.3 |
| hsa-miR-101-3p | -0.582 | 1.19E-02 | 1.81E-01 | 2085.32 | 3118.13 |

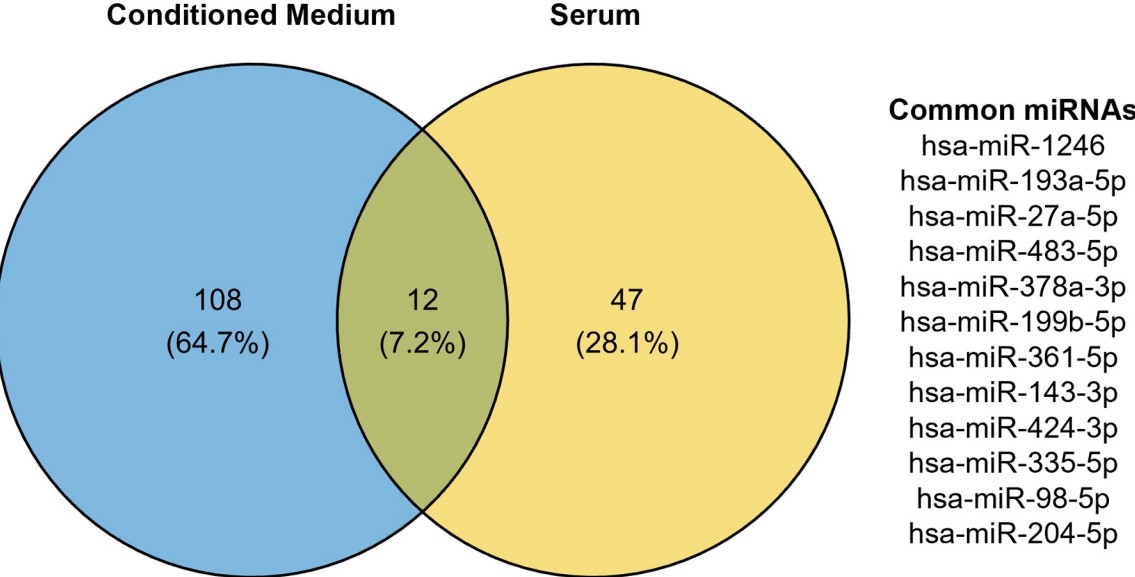

**Common miRNAs**
hsa-miR-1246
hsa-miR-193a-5p
hsa-miR-27a-5p
hsa-miR-483-5p
hsa-miR-378a-3p
hsa-miR-199b-5p
hsa-miR-361-5p
hsa-miR-143-3p
hsa-miR-424-3p
hsa-miR-335-5p
hsa-miR-98-5p
hsa-miR-204-5p

**Fig 3. Venn diagram identifying common differentially expressed miRNA in day 7 conditioned medium from osteogenic cultures and day 3 serum samples from fracture patients.**

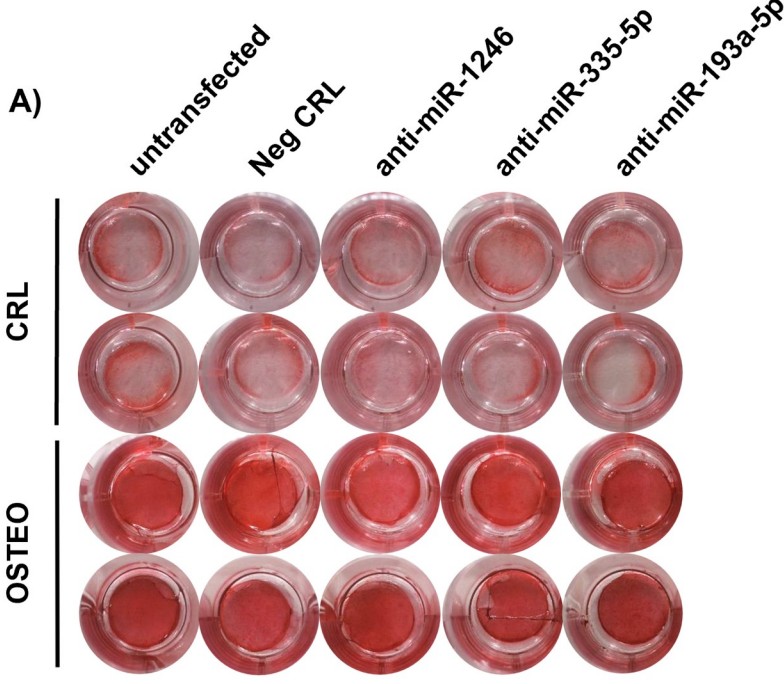

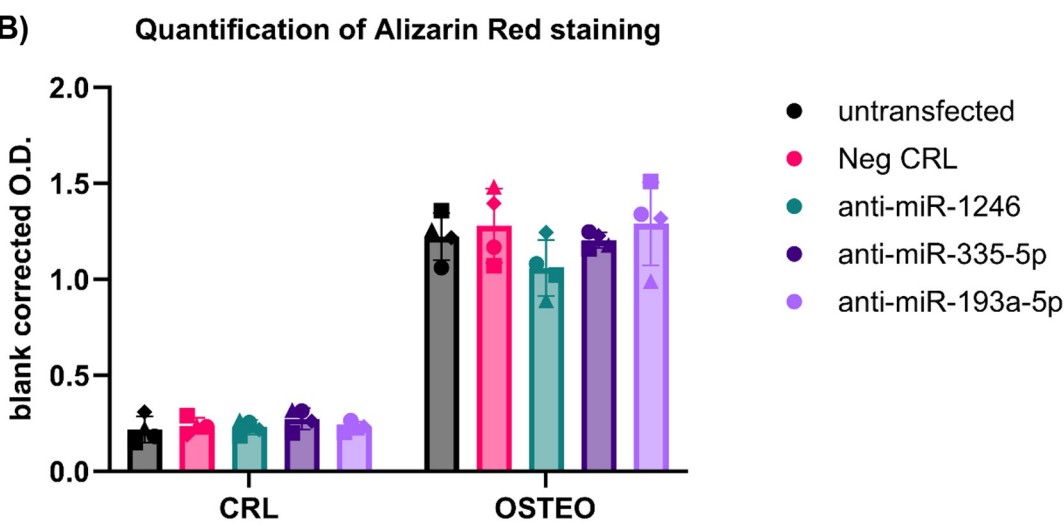

**Fig 4. Analysis of mineral deposition after treatment of BMSCs with miRNA inhibitors.** A) Representative macroscopic overview of Alizarin Red staining from one donor. B) Alizarin Red quantification from n = 4 donors.

## Conclusions

miR-1246, miR-335-5p, and miR-193a-5p can be involved in fracture healing, though they do not have a role in direct osteogenesis of human BMSCs *in vitro*. However, direct ossification is not the only or the main mechanism going on during fracture healing. Therefore, it is possible

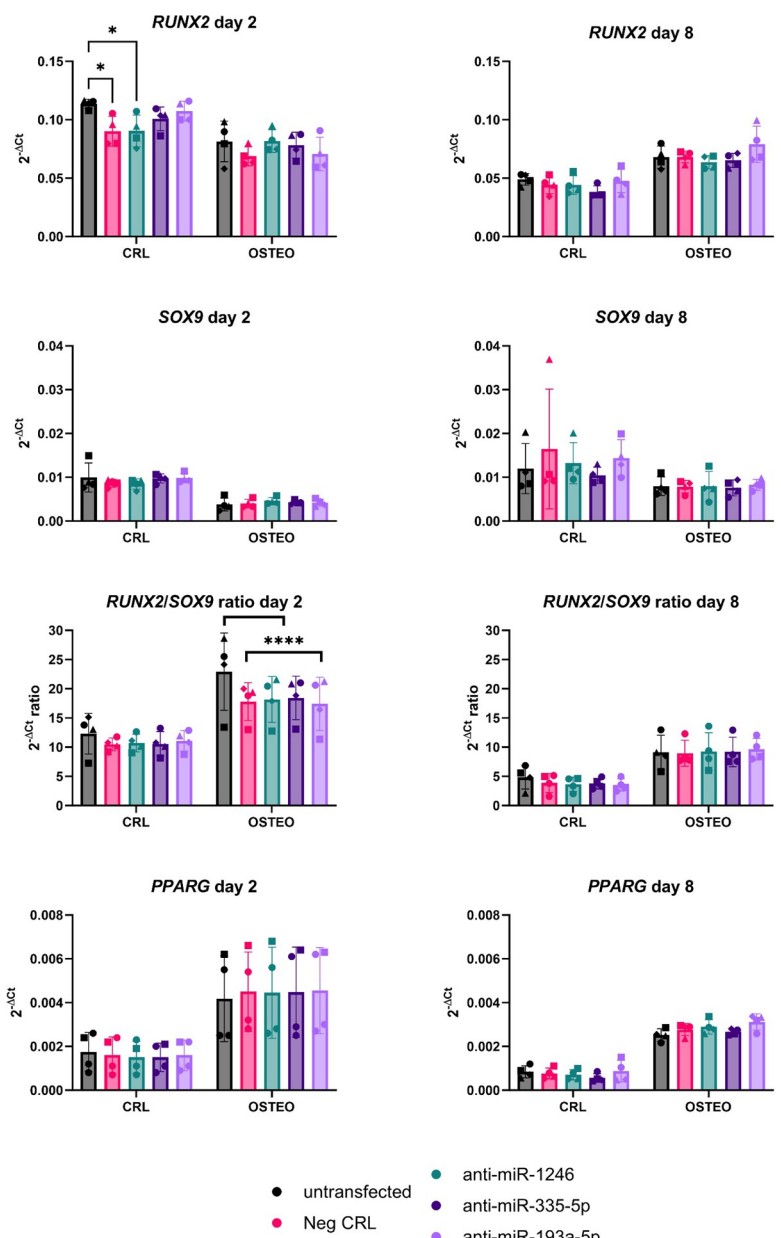

**Fig 5. Gene expression analysis of cell fate markers after treatment of BMSCs with miRNA inhibitors.** Left: day 2 after transfection; right: day 8 after transfection. Data are reported as $2^{-\Delta Ct}$, with normalization to the reference gene *RPLP0*. A repeated measure-two-way ANOVA with Tukey's multiple comparison test was used to compare the mean differences between the groups. * $p<0.05$; **** $p>0.0001$. For all the genes, differentiation (i.e., CRL vs. OSTEO) is the main factor inducting changes in gene expression.

that those miRNAs can be involved in other key processes such as cell recruitment, vascularization, or endochondral ossification. miR-1246 seems the most promising to be further characterized, as it is mechanosensitive, promotes proliferation and its expression can be regulated by key chemokines involved in MSC recruitment and bone formation.

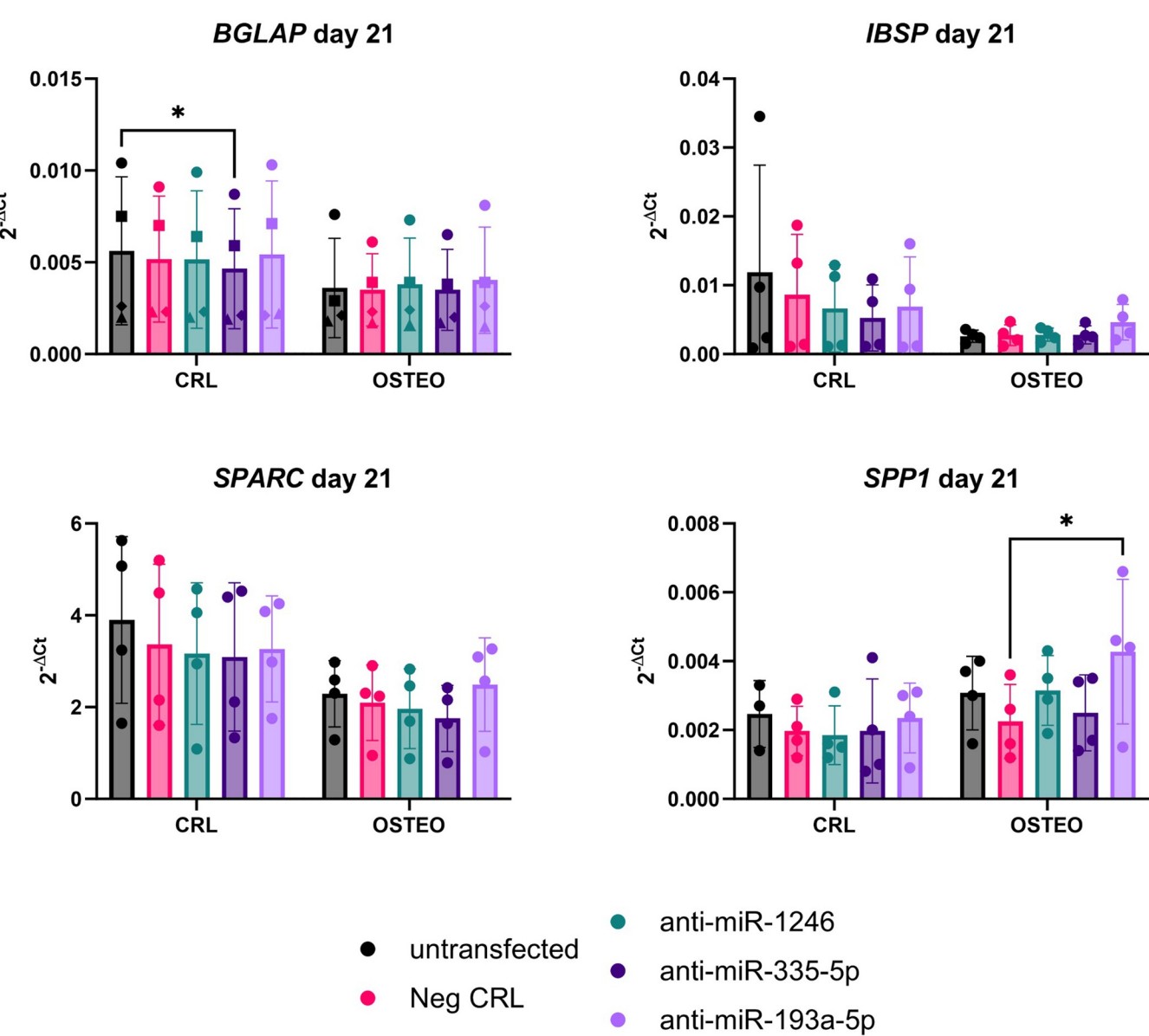

**Fig 6. Gene expression analysis of late osteogenic differentiation markers after treatment of BMSCs with miRNA inhibitors.** Data are reported as $2^{-\Delta Ct}$, with normalization to the reference gene *RPLP0*. A repeated measure-two-way ANOVA with Tukey's multiple comparison test was used to compare the mean differences between the groups. * $p<0.05$.

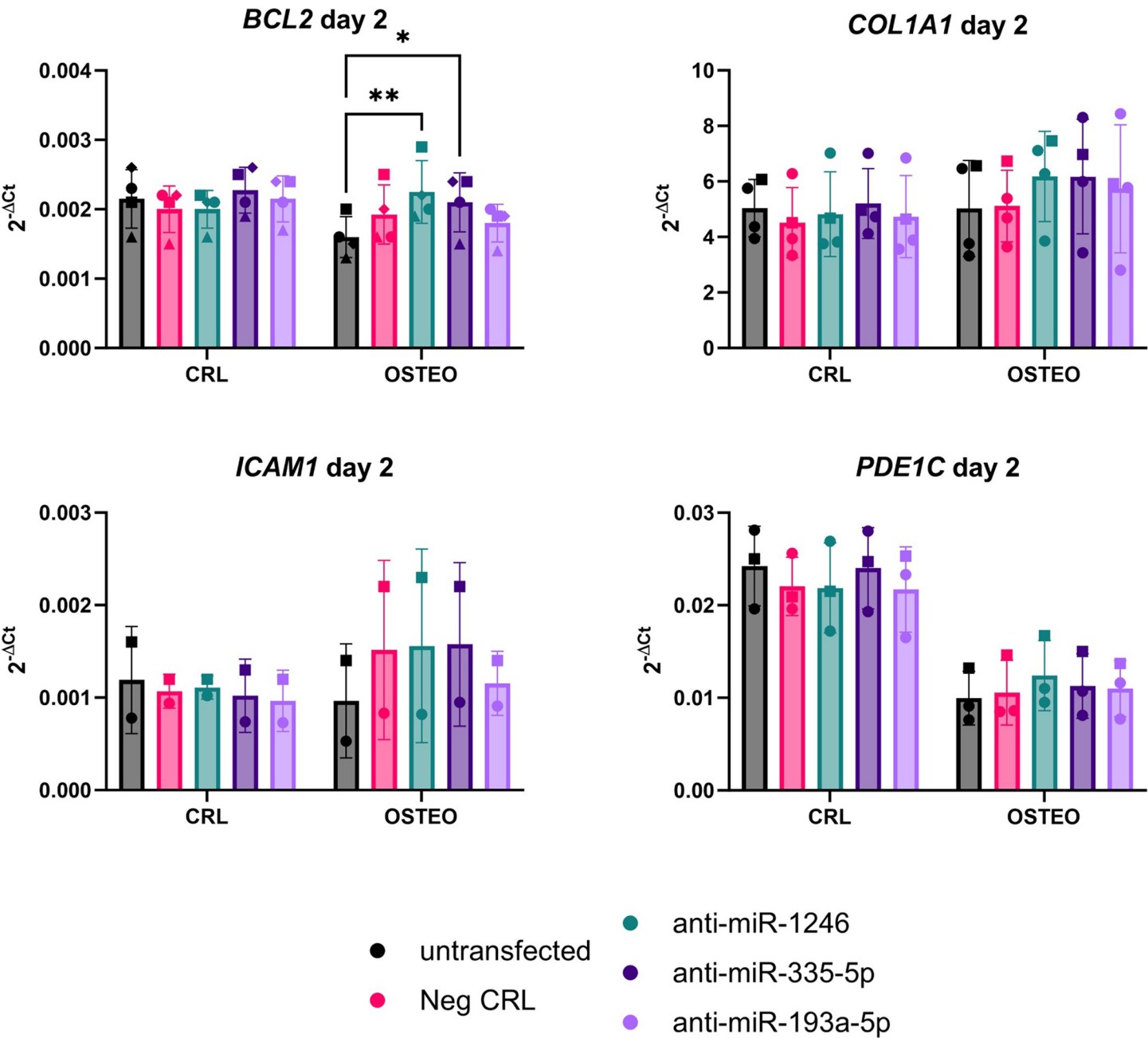

**Fig 7. Gene expression analysis of putative miRNA targets two days after transfection with the miRNA inhibitors.** Data are reported as $2^{-\Delta Ct}$, with normalization to the reference gene *RPLP0*. A repeated measure-two-way ANOVA with Tukey's multiple comparison test was used to compare the mean differences between the groups. * $p<0.05$. ** $p<0.01$.

## Supporting information

**S1 Table. Summary of subjects enrolled in the study.**
(DOCX)

**S2 Table. qPCR assay details.**
(DOCX)

**S3 Table. Full list of differentially regulated miRNA at day 7 in conditioned medium (osteo vs ctrl), sorted by Log2FC.**
(DOCX)

**S4 Table. Full list of differentially regulated miRNA at day 15 in conditioned medium (osteo vs ctrl), sorted by Log2FC.**
(DOCX)

**S1 Fig. Analysis of differentially expressed miRNA in conditioned medium (day 14 osteo vs. ctrl).** In volcano plots, the x-axis reports the log2 fold-change between osteo and ctrl (Log2FC), while the y-axis represents the -log10 of the p-value. Thresholds are set at 0.58 of log2FC (corresponding to an absolute 1.5-fold-change value) and to a -log10 p-value of 1.3 (corresponding to p-value of 0.05).
(PDF)

**S2 Fig. Volcano plot identifying differentially expressed miRNAs in serum of fracture patients.** A) day 3 fracture serum samples versus controls. B) day 5–12 fracture serum samples versus controls. C) day 19–56 fracture serum samples versus controls.
(PDF)

**S3 Fig. Analysis of cellular miRNA expression during osteogenic differentiation, at day 2 and at day 21.** *p<0.05. **p<0.01.
(PDF)

**S1 Text.**
(TXT)

**S1 Dataset.**
(XLSX)

## Author Contributions

**Conceptualization:** Elena Della Bella, Martin J. Stoddart.

**Data curation:** Elena Della Bella, Ursula Menzel, Andreas Naros, Eva Johanna Kubosch.

**Formal analysis:** Elena Della Bella, Ursula Menzel, Andreas Naros, Mauro Alini, Martin J. Stoddart.

**Funding acquisition:** Elena Della Bella, Martin J. Stoddart.

**Investigation:** Ursula Menzel, Andreas Naros, Eva Johanna Kubosch, Mauro Alini.

**Methodology:** Elena Della Bella, Ursula Menzel, Eva Johanna Kubosch.

**Project administration:** Andreas Naros, Eva Johanna Kubosch, Mauro Alini, Martin J. Stoddart.

**Resources:** Eva Johanna Kubosch, Mauro Alini.

**Supervision:** Elena Della Bella, Ursula Menzel, Mauro Alini, Martin J. Stoddart.

**Writing – original draft:** Elena Della Bella, Martin J. Stoddart.

**Writing – review & editing:** Elena Della Bella, Andreas Naros, Eva Johanna Kubosch, Mauro Alini, Martin J. Stoddart.

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
