## [Decision Letter · Decision Letter 0]

26 Jun 2023

PONE-D-23-12866Identification of circulating miRNAs as fracture-related biomarkers .PLOS ONE

Dear Dr. Stoddart,

Thank you for submitting your manuscript to PLOS ONE. After careful consideration, we feel that it has merit but does not fully meet PLOS ONE’s publication criteria as it currently stands. Therefore, we invite you to submit a revised version of the manuscript that addresses the points raised during the review process. Please submit your revised manuscript by Aug 10 2023 11:59PM. If you will need more time than this to complete your revisions, please reply to this message or contact the journal office at plosone@plos.org. Please include the following items when submitting your revised manuscript:A rebuttal letter that responds to each point raised by the academic editor and reviewer(s). You should upload this letter as a separate file labeled 'Response to Reviewers'.A marked-up copy of your manuscript that highlights changes made to the original version. You should upload this as a separate file labeled 'Revised Manuscript with Track Changes'.An unmarked version of your revised paper without tracked changes. You should upload this as a separate file labeled 'Manuscript'.

We look forward to receiving your revised manuscript.

Kind regards,

Carlos Alberto Antunes Viegas, DVM; MSc; PhD

Academic Editor

PLOS ONE

Journal Requirements:

Reviewers' comments:

Reviewer's Responses to Questions

**Comments to the Author**

1. Is the manuscript technically sound, and do the data support the conclusions?

Reviewer #1: Partly

Reviewer #2: Yes

2. Has the statistical analysis been performed appropriately and rigorously? 

Reviewer #1: I Don't Know

Reviewer #2: Yes

3. Have the authors made all data underlying the findings in their manuscript fully available?

Reviewer #1: Yes

Reviewer #2: Yes

4. Is the manuscript presented in an intelligible fashion and written in standard English?

Reviewer #1: No

Reviewer #2: Yes

5. Review Comments to the Author

Reviewer #1: Manuscript ID: PONE-D-23-12866

Title: Identification of circulating miRNAs as fracture-related biomarkers.

Bella et al., studied the potential role of serum miRNA as biomarkers for bone fractures. Three miRNAs were identified and assessed for their effects on osteogenic differentiation. Below are some comments/suggestions to be considered.

• Were blood samples collected at different stages collected from same patients? For instance, did blood samples collected at early, late and very late stages belong to same patient, at least in some of them?

• Were human BMSCs collected from different patients pooled together for the differentiation experiments? And why did authors specifically use male donor? Although fracture patients’ samples included males and females

• What are the exclusion criteria for all selected patients and donors used for the study?

• A reference for the used method of osteogenic differentiation should be included.

• It is not cleared whether the conditioned media was serum free or not. How many times did authors change media during differentiation process?

• miRNA inhibitor transfection is a transient process, authors mentioned collecting the cells 8- and 21-days post transfection. Did authors verify miRNA inhibition at those time points?

• The article needs revision by a biostatistician.

Reviewer #2: Elena Della Bella et al. identified the miR-1246 may be involved in cell proliferation and recruitment of progenitor cells. This study give insight into understanding of miRNAs as fracture-related biomarkers. I have some major concerns:

1. Although this manuscript is well prepared, the language should be polished by a native speaker at deep extent.

2. Text and Figures could be refined and reorganized, for example, fonts size in figures should be identical.

3. The Introduction section should be re-arranged and refined to clarify the role of miRNAs in fracture and bone disease.

4. Since osteogenic differentiation is not the main contributor to fracture healing, it is interesting to analyze whether miR-1246 is involved in osteoporosis in patient samples or mouse model.

5. Why hsa26 miR-1246, hsa-miR-335-5p, and miR-193a-5p do not involved in BMSC osteogenic differentiation should be discussed.

6. Role of miR-1246 in BMSCs osteogenic differentiation should be further detected, for example, Correlation of BGLAP, OPN, COL1A1, RUNX2 and miR-1246 expression in BMSCs should be analyzed, alkaline phosphatase assay (ALP) staining should be performed.

7. The authors identified miRNA changes in fracture patients versus healthy controls and correlated those changes with differential miRNA expression during in vitro osteogenic differentiation, does MSCs from fracture patients and healthy controls show difference in their abilities of osteogenic differentiation?

6. PLOS authors have the option to publish the peer review history of their article (what does this mean?). If published, this will include your full peer review and any attached files.

Reviewer #1: No

Reviewer #2: No

---

## [Author Response · Author response to Decision Letter 0]

8 Jan 2024

Reviewer #1:

Manuscript ID: PONE-D-23-12866

Title: Identification of circulating miRNAs as fracture-related biomarkers.

Bella et al., studied the potential role of serum miRNA as biomarkers for bone fractures. Three miRNAs were identified and assessed for their effects on osteogenic differentiation. Below are some comments/suggestions to be considered.

• Were blood samples collected at different stages collected from same patients? For instance, did blood samples collected at early, late and very late stages belong to same patient, at least in some of them?

Table S1 in the supplementary information file contains this information. As evidenced, for some donors we had samples from multiple time points, while for some others just one time point. Therefore, we consider this study as cross sectional and we did not analyse the differences in miRNA in time from the same donor, when multiple serum samples were collected. This is also briefly discussed within the study limitations.

• Were human BMSCs collected from different patients pooled together for the differentiation experiments? And why did authors specifically use male donor? Although fracture patients’ samples included males and females

Samples from different donors were not pooled but used separately. All male donors were used as although none of the identified markers indicated a sex-based difference, we attempted to remove as many confounders as possible. The male only gender differentiation results are already mentioned as a limitation of the study. 

• What are the exclusion criteria for all selected patients and donors used for the study?

Exclusion criteria for fracture patients included open fractures, HIV, hepatitis, smokers, diabetics, obese patients. For BMSC donors, no exclusion criteria were applied, and the donors were fully anonymised. 

• A reference for the used method of osteogenic differentiation should be included.

A reference to the method used for osteogenic differentiation was added: Gardner OFW, Alini M, Stoddart MJ. Mesenchymal Stem Cells Derived from Human Bone Marrow. In: Doran PM, editor. Cartilage Tissue Engineering: Methods and Protocols. New York, NY: Springer New York; 2015. p. 41-52. This is reference #19 in the revised manuscript.

• It is not cleared whether the conditioned media was serum free or not. How many times did authors change media during differentiation process?

As stated in the materials and methods (lines 94-98 of the revised manuscript) cells were cultured in serum-free conditions for 48 hours before the collection of the conditioned medium. Medium was refreshed three times per week throughout all the differentiation process. This information is added to the revised manuscript, line 98. 

• miRNA inhibitor transfection is a transient process, authors mentioned collecting the cells 8- and 21-days post transfection. Did authors verify miRNA inhibition at those time points?

The transfection of miRNA inhibitors is indeed a transient process, with antisense effects normally assessed 24–72 hours after transfection. We performed a single transfection the day after cell seeding and collected samples 2, 8, and 21 days after the delivery of miRNA inhibitors. miRNA inhibition cannot be evaluated by analysing the expression levels of the miRNA themselves, as inhibitors form stable complexes with their target, but do not induce their degradation. Therefore, the expression of miRNA targets should be evaluated as a measure of their inhibition. To this purpose, Figure 7 focuses on the analysis of putative miRNA targets at day 2. However, multiple papers have demonstrated that short term inhibition of relevant targets have long term effects e.g. DOI: 10.3389/fbioe.2020.00618. 

• The article needs revision by a biostatistician.

We have checked the statistics and are confident they are valid.

Reviewer #2: Elena Della Bella et al. identified the miR-1246 may be involved in cell proliferation and recruitment of progenitor cells. This study give insight into understanding of miRNAs as fracture-related biomarkers. I have some major concerns:

1. Although this manuscript is well prepared, the language should be polished by a native speaker at deep extent.

We have checked the manuscript for English and made the required corrections.

2. Text and Figures could be refined and reorganized, for example, fonts size in figures should be identical.

The figures were revised accordingly.

3. The Introduction section should be re-arranged and refined to clarify the role of miRNAs in fracture and bone disease.

The introduction was re-arranged according to the Reviewer’s suggestions (lines 43-64 of the revised manuscript). 

4. Since osteogenic differentiation is not the main contributor to fracture healing, it is interesting to analyze whether miR-1246 is involved in osteoporosis in patient samples or mouse model.

We thank the reviewer for this comment. The literature regarding the role of miR-1246 in osteoporosis or in bone remodelling is limited, however there are a few papers suggesting it as a regulator of osteoblast and osteoclast formation. Zhou et al (10.1016/j.diff.2020.06.004) reported that miR-1246 in FBS-derived exosomes can attenuate adipogenic differentiation of human BM-MSCs, therefore having an indirect positive role on bone formation. Nguyen et al (10.1007/s00109-021-02128-5) found miR-1246 as significantly reduced in pagetic overactive osteoclasts. In another study, miR-1246 was the most upregulated miRNA in circulating EVs from osteoporotic patients compared to those from healthy controls (10.1002/jbmr.4688), and the treatment of osteoclast precursors with miR-1246 mimic enhanced osteoclastogenesis (in accordance with Liao et al 10.4161/cc.20809 which reported the activation of NFATC1 downstream of mir-1246), but also upregulated SP7 in osteoblasts, indicating a role in bone remodelling processes. However, the same authors observed a decreased expression of this miRNA in osteoblasts compared to undifferentiated MSCs. This, together with our data from differentiating MSCs, might suggest that miR-1246 expression peaks during the pre-osteoblast stage. 

This part was added to the discussion on the revised manuscript (lines 318-330).

5. Why hsa26 miR-1246, hsa-miR-335-5p, and miR-193a-5p do not involved in BMSC osteogenic differentiation should be discussed.

We thank the reviewer for giving us the opportunity to clarify this point in the discussion. We added the following to the discussion (lines 353-358), before the study limitations, which include also other considerations on miRNA role in other fracture repair-related mechanisms. 

“Altogether, our results suggest that miR-1246, miR-193a-5p, and miR-335-5p are detectable in the serum of fracture patients with reproducible patterns of expression and show a differential expression during in vitro osteogenic differentiation of human BMSC, but they are not directly causative of direct ossification. Therefore, we think that this miRNA pattern reflects a downstream change, rather than being an upstream regulator of hBMSC direct differentiation. Notwithstanding, we cannot exclude that these miRNAs have a functional role in other fracture healing-related processes.”

6. Role of miR-1246 in BMSCs osteogenic differentiation should be further detected, for example, Correlation of BGLAP, OPN, COL1A1, RUNX2 and miR-1246 expression in BMSCs should be analyzed, alkaline phosphatase assay (ALP) staining should be performed.

We thank the reviewer for this suggestion. Accordingly, we ran additional qPCRs and analysed the expression of miR-1246, miR-335-5p, and miR-193a-5p at different timepoints in non-transfected cells. The expression of miR-1246 was found to be non-significantly reduced at 2 days, while it was significantly upregulated at 21 days of osteogenic differentiation, compared to the controls. miR-335-5p was significantly downregulated in osteogenic differentiation at both timepoints, while miR-193a-5p did not show any significant difference between conditions. 

Moreover, we ran correlations between miRNA expression and late markers of osteogenic differentiation. The levels of miR-1246, miR-335-5p and miR-193a-5p at any timepoint did not show any significant correlation to other osteogenic marker (data not shown).

These results further support our conclusions, confirming that these miRNAs have no direct functional role in direct osteogenic differentiation of MSCs, but their regulation is likely to reflect a downstream change in the differentiation process.

New methods and results data about miRNA expression and correlations throughout the differentiation process was added to the manuscript (lines 145-148, 248-254, 258-260, 560-561, and Supplementary Figure 3).

7. The authors identified miRNA changes in fracture patients versus healthy controls and correlated those changes with differential miRNA expression during in vitro osteogenic differentiation, does MSCs from fracture patients and healthy controls show difference in their abilities of osteogenic differentiation?

In this work we did not assess the potential for osteogenic differentiation of MSCs from healthy vs fracture patients. Our BMSC donors are all fracture patients. However, we expect the MSC potential to be mainly influenced by factors such as diabetes, smoking, obesity, alcohol consumption, etc. (see, e.g., 10.1002/sctm.19-0044; 10.1097/HS9.0000000000000615].

---

## [Decision Letter · Decision Letter 1]

18 Apr 2024

Identification of circulating miRNAs as fracture-related biomarkers .

PONE-D-23-12866R1

Dear Dr. Martin J Stoddart,

We’re pleased to inform you that your manuscript has been judged scientifically suitable for publication and will be formally accepted for publication once it meets all outstanding technical requirements.

Kind regards,

Carlos Alberto Antunes Viegas, DVM; MSc; PhD

Academic Editor

PLOS ONE

Additional Editor Comments (optional):

Reviewers' comments:

Reviewer's Responses to Questions

**Comments to the Author**

1. If the authors have adequately addressed your comments raised in a previous round of review and you feel that this manuscript is now acceptable for publication, you may indicate that here to bypass the “Comments to the Author” section, enter your conflict of interest statement in the “Confidential to Editor” section, and submit your "Accept" recommendation.

Reviewer #1: All comments have been addressed

Reviewer #2: (No Response)

2. Is the manuscript technically sound, and do the data support the conclusions?

Reviewer #1: Yes

Reviewer #2: Partly

3. Has the statistical analysis been performed appropriately and rigorously? 

Reviewer #1: (No Response)

Reviewer #2: Yes

4. Have the authors made all data underlying the findings in their manuscript fully available?

Reviewer #1: (No Response)

Reviewer #2: Yes

5. Is the manuscript presented in an intelligible fashion and written in standard English?

Reviewer #1: (No Response)

Reviewer #2: Yes

6. Review Comments to the Author

Reviewer #1: (No Response)

Reviewer #2: Major concerns not resolved. Some experimets should be added to support the conclusion and make the conclusion more convincable.

7. PLOS authors have the option to publish the peer review history of their article (what does this mean?). If published, this will include your full peer review and any attached files.

Reviewer #1: No

Reviewer #2: **Yes: **Jingya Wang

---

## [Editor Report · Acceptance letter]

29 Apr 2024

PONE-D-23-12866R1 

PLOS ONE

Dear Dr. Stoddart, 

I'm pleased to inform you that your manuscript has been deemed suitable for publication in PLOS ONE. Congratulations! Your manuscript is now being handed over to our production team.

Kind regards, 

on behalf of

Dr. Carlos Alberto Antunes Viegas 

Academic Editor

PLOS ONE